

# Reliability and responsiveness of a tissue hardness meter and algometer for measuring tissue hardness and pressure pain threshold in upper trapezius myofascial trigger points

Soukmisai Somphithak[1], Uraiwan Chatchawan[1,2], Atipong Pimdee[1,2] and Wiraphong Sucharit[1,2]

[1] School of Physical Therapy, Faculty of Associated Medical Sciences (AMS), Khon Kaen University (KKU), Khon Kaen, Thailand
[2] Research Center in Back, Neck, Other Joint Pain and Human Performance (BNOJPH), Khon Kaen University (KKU), Khon Kaen, Thailand

## ABSTRACT

**Background:** Tissue hardness meter and algometer (THA) are used to assess tissue hardness (TH) and pressure pain threshold (PPT), particularly in the evaluation of myofascial trigger points (MTrPs). This study introduces a side-lying protocol designed to comprehensively measure all portions of the upper trapezius (UT) muscle.

**Purpose:** The objective was to determine the reliability and responsiveness of THA to measure TH and PPT in patients with MTrPs in the UT muscle.

**Methods:** Reliability of TH and PPT measurements was assessed in a sample of 24 participants. Intra-rater and inter-rater reliability were evaluated using the intra-class correlation coefficient ($ICC_{3,1}$), while absolute reliability was established *via* Bland–Altman analysis, including the calculation of 95% limits of agreement (95% LoA). To assess responsiveness, 36 additional participants were recruited. Both distribution-based methods (mean difference, effect size (ES), standardized response mean (SRM), standard error of measurement (SEM), and minimal detectable change at 95% confidence ($MDC_{95}$)) and anchor-based methods (minimal clinically important difference (MCID) and area under the curve (AUC)) were utilized in the analysis.

**Results:** Intra-rater reliability was excellent for both TH and PPT ($ICC_{3,1}$: 0.95–0.97), while inter-rater reliability was moderate ($ICC_{3,1}$: 0.60). Evidence of both fixed and proportional bias was identified for both TH and PPT. For TH, the SEM and $MDC_{95}$ were 2.66% and 7.37%, respectively, while for PPT, they were 0.12 kg/cm$^2$ and 0.34 kg/cm$^2$, respectively. Following six physical therapy sessions, significant reductions in TH (mean: −7.86%; MCID: −7.43%; AUC: 0.97) and significant increases in PPT (mean: 0.20 kg/cm$^2$; MCID: 0.21 kg/cm$^2$; AUC: 0.86) were observed. Additionally, changes in PPT showed a negative correlation with improvements in the Neck Disability Index (NDI) (r = −0.35, $p < 0.05$).

**Conclusion:** The side-lying protocol demonstrated reliable and clinically relevant TH and PPT measurements, supporting its use for monitoring treatment outcomes in patients with MTrPs in the UT muscle.

Corresponding author
Wiraphong Sucharit,
wirasu@kku.ac.th

# INTRODUCTION

The tissue hardness meter and algometer (THA) were developed to measure of both tissue hardness (TH) and pressure pain threshold (PPT) (*Morozumi et al., 2010*). It is widely used for patients with myofascial trigger points (MTrPs), which are sensitive nodules within taut bands of muscle fibers (*Simons & Travell, 1999*). These trigger points are a primary cause of chronic pain and disability. Measurements of TH and PPT in MTrPs using the THA have shown increased TH (*Sikdar et al., 2009*) and reduced PPT (*Park et al., 2011*). The THA operates based on specific principles for measuring TH and PPT. For accurate measurement, the device's probe must be positioned perpendicularly to the target area, directly influencing patient positioning. Previous studies have predominantly used prone or supine positions (*Morozumi et al., 2022*). In measuring TH, the device utilizes two sensors: the outer sensor applies a controlled force of 30 N to compress the tissue, while the inner sensor simultaneously measures the resulting displacement. Once the preset force is reached, the device calculates TH by converting the indentation depth into a percentage relative to a reference tissue. For PPT measurements, the rater gradually increases the pressure until the patient feels pain and presses a button to stop the test, at which point the device records the pressure (*Morozumi et al., 2010*). This tool is convenient, making it an asset in clinical practice for assessing the effectiveness of treatment interventions (*Damapong et al., 2015*; *Buttagat et al., 2021*; *Huang et al., 2022*; *Sucharit et al., 2023*).

Although previous studies have demonstrated that the THA is a valid and reliable tool for measuring TH and PPT (*Morozumi et al., 2010*), variability in measurements persists. This variability is often attributed to differences in the experience level of the individual conducting the assessment and the condition of the muscle being evaluated (*Morozumi et al., 2022*). Additionally, inconsistencies in pressure application and the speed of force can affect measurement reliability (*Kinser, Sands & Stone, 2009*). Establishing consistent methods is essential for both clinical practice and research. The THA has been widely used to study the effects of treatment interventions on MTrPs in the upper trapezius (UT) muscle (*Damapong et al., 2015*; *Buttagat et al., 2021*; *Sucharit et al., 2023*). However, many studies lack detailed protocols (*Buttagat, Eungpinichpong & Chatchawan, 2011*; *Buttagat et al., 2012*, *2021*; *Damapong et al., 2015*; *Areeudomwong et al., 2022*; *Buttagat, Kluayhomthong & Areeudomwong, 2023*), particularly regarding patient positioning. For instance, when participants are positioned in a supine position, there may be limitations in measuring the posterior portion of the UT muscle, while a prone position may restrict access to the anterior portion. Differences in patient positioning may affect the condition of the muscle being assessed (*Morozumi et al., 2022*), potentially serving as a confounding factor in the TH and PPT measurements.

To address these limitations, we developed a protocol in which patients are positioned on their side. This position enables consistent measurements of the anterior, middle, and posterior parts of the UT muscle without changing the patient's position. It also ensures

adequate support for the head, neck, shoulder, and arm, which may promote muscle relaxation—an essential factor in reducing variability in both TH and PPT measurements. Since muscle tone can influence TH values, and uneven pressure distribution may affect pain perception (*Somprasong et al., 2015*), thereby impacting the reliability of the PPT measurement, maintaining a relaxed muscle state is critical for improving the consistency of both assessments.

We hypothesize that the newly developed side-lying protocol will enhance the intra-rater and inter-rater reliability of TH and PPT measurements in individuals with MTrPs in the UT muscle. Furthermore, this approach may enhance responsiveness to changes in TH and PPT values. To test this hypothesis, we aimed to (1) evaluate the intra- and inter-rater reliability of TH and PPT measurements using a standardized protocol, and (2) assess their responsiveness to clinical change following physical therapy intervention. We further explored the correlation between changes in objective measurements and patient-reported outcomes such as pain intensity and Neck Disability Index.

## METHODS

### Design and setting

This study was conducted in the Department of Physical Therapy, Faculty of Associated Medical Science (AMS), Khon Kaen University (KKU), Thailand. It was approved by the Research Ethics Committee of KKU in accordance with the Declaration of Helsinki (Reference: HE662181) and is registered in the Thai Clinical Trails Registry (TCTR20240618009). All participants provided written informed consent prior to their involvement in the study.

### Participants

Twenty-four participants were recruited to assess the reliability by both expert and beginner raters. Additionally, 36 participants were recruited to evaluate the responsiveness of TH and PPT following six physical therapy (PT) sessions conducted over a two-week period. Participants were recruited over a 3-month period (June to August 2024) through advertisements at a government clinic and other local service departments. A physical therapist with 10 years of experience conducted screening using questionnaires and physical examinations. Inclusion criteria required participants to be aged 18–40 years, residing in Khon Kaen province, and diagnosed with active MTrPs in at least one side of the UT muscle, based on standard diagnostic criteria (*Simons, 2008*). Exclusion criteria included a body mass index (BMI) $\geq 30$ kg/m$^2$, uncontrolled vital signs, complete disability (measured by the NDI), cervical myelopathy or radiculopathy (*Valera-Calero et al., 2021*), diabetes mellitus, or any recent surgical procedures involving the head or shoulder region.

Sample size was determined for testing the measured reliability using a hypothesis ICC value ($P_1$) of 0.75, a null value ($P_0$) of 0.5, and three repeated measurements. To achieve 80% power at a 5% level of significance, 24 participants were recruited. To assess whether changes in TH and PPT reflect clinical improvements, another sample size was estimated based on a pilot study of six participants, where the mean TH decreased from $48.25 \pm 7.27\%$ before treatment to $40.67 \pm 6.03\%$ after six sessions of PT treatment. Assuming a

50% reduction in the mean difference observed in the pilot study ($-7.6 \pm 7.20\%$) and accounting for a 10% dropout rate, a total of 36 participants were recruited.

## Intervention

Participants received standard physical therapy for the neck and shoulder from the physical therapist. The therapy included ultrasound treatment (1.5 W/cm$^2$ intensity, 1 MHz frequency for 5 min) (*Yildirim, Öneş & Gökşenoğlu, 2018*), passive stretching of the UT muscle (30 s per stretch, 2 times per side) (*Simons & Travell, 1999*), and hot pack application (70–75 °C) for 15 min (*Benjaboonyanupap, Paungmali & Pirunsan, 2015*). Each session lasted 30 min and was provided 3 times per week for 2 weeks.

## Outcome measures

### Side-lying positioning protocol for TH and PPT measurement

Participants were positioned in a relaxed side-lying position on a firm treatment table, with the painful side facing upward. The head and neck were supported by an adjustable pillow to maintain a neutral cervical alignment, and participants were instructed to avoid head rotation during the measurements. The ipsilateral (upper) shoulder and arm were comfortably supported using an additional adjustable pillow to promote muscle relaxation. The upper limb was positioned with slight shoulder adduction and mild flexion, while the elbow was maintained in a mid-range flexed position. Meanwhile, the contralateral (lower) shoulder was placed in slight protraction, flexion, and external rotation to enhance postural stability. The ipsilateral hip and knee were slightly flexed, and the contralateral lower limb was flexed at both the hip and knee and supported with a firm pillow to help maintain lumbar neutrality (Fig. 1).

## TH and PPT measurements

TH and PPT were measured using a dual-function tissue hardness meter and algometer (THA) (OE-220; ITO Co., Ltd., Tokyo, Japan) (*Morozumi et al., 2010*), which has demonstrated high reliability and accuracy in previous validation studies. The reported ICC ranged from 0.86 to 0.96 for TH measurements under varying loads, confirming the device's precision. Additionally, the algometer function yielded ICC of 0.95 for pain threshold and 0.94 for pain tolerance, indicating excellent reliability (*Morozumi et al., 2010*).

For TH, a flat plastic disk (10 cm in diameter) was placed perpendicular to the skin over the identified MTrP (Fig. 1A). The rater held the THA device with both hands and applied steady, gradual pressure against the measurement site. The sensor automatically detected when a constant force was maintained for 3 s, at which point an audible beep confirmed successful data capture. TH was expressed as a percentage (%) (*Morozumi et al., 2010*). For PPT, the disk was replaced with a 1 cm$^2$ rubber tip, which was positioned perpendicularly over the same MTrP previously identified for the TH measurement. Pressure was gradually applied at a constant rate of approximately 1 kg/cm$^2$ per second until the participant began to feel pain and pressed an algometer switch. The PPT was recorded in kilograms per square centimeter (kg/cm$^2$) (*Nussbaum & Downes, 1998*) (Fig. 1B).

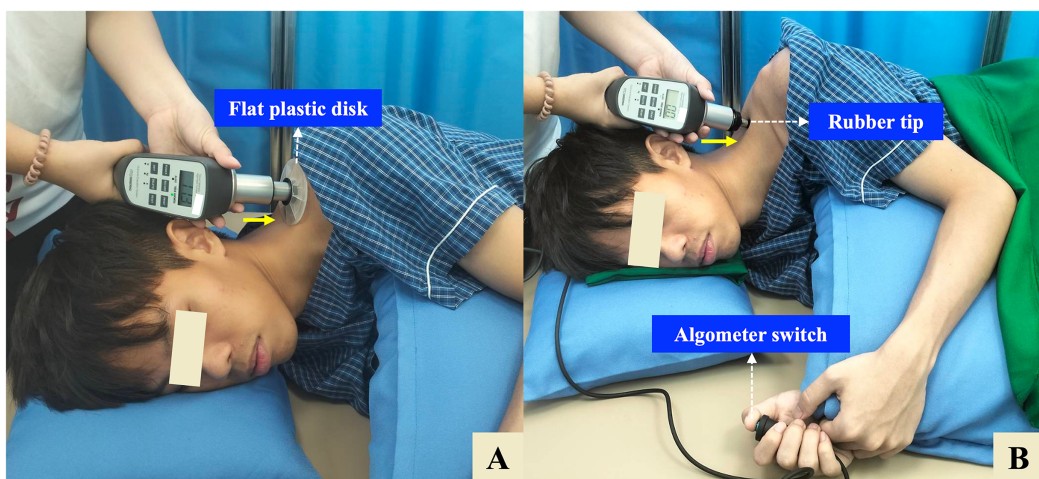

**Figure 1** **Side-lying protocol for measuring tissue hardness (TH) and pressure pain threshold (PPT) using a combined tissue hardness and algometer (THA) device.** (A) TH measurement: A flat plastic disk (10 cm diameter) is applied perpendicularly over the identified myofascial trigger point (MTrP) on the upper trapezius (UT) muscle. (B) PPT measurement: The disk is replaced with a 1-cm$^2$ rubber tip, and pressure is gradually applied until the participant presses the algometer switch to indicate the onset of pain. The solid yellow arrow indicates the direction of pressure, which is applied approximately perpendicular (90°) to the MTrP. The participant is positioned in a relaxed side-lying posture with full support for the head, neck, shoulder, and arm to ensure muscle relaxation.

## Clinical outcome measurements

Pain intensity was assessed using a numerical visual analog scale (VAS), where 0 cm indicated "no pain" and 10 cm indicated the "worst pain imaginable" (*Carlsson, 1983*). Neck disability was measured using the Neck Disability Index, Thai version (NDI-Thai Version) (*Uthaikhup, Paungmali & Pirunsan, 2011*), with participants scoring limitations across 10 items from 0 (no limitation) to 50 (complete limitation). The Global Rating of Change (GRC) was measured on a 15-point Likert scale, ranging from −7 (worsening) to +7 (improvement), with 0 representing no change (*Kamper, Maher & Mackay, 2009*).

## Procedure

Before participant recruitment, the beginner rater received 20 h of practice. A calibration block with known hardness (100%) was used; pressure was applied until the outer sensor reached 30 N, after which the inner sensor was expected to show 100%. If it did not, a full professional calibration was performed by the manufacturer. For reliability assessment, two raters (expert and beginner) were involved. The order of measurements was randomized for each of the 24 participants using computer-generated numbers. To minimize observer bias, the beginner rater and the expert rater were blinded to each other's results. All measurements were conducted and recorded independently. The physiotherapist identified the most painful MTrP in the UT muscle through manual palpation and marked the location on the skin using permanent ink. Participants then rested for 5 min prior to the measurements. TH and PPT were each measured three times per rater, with a 2-min interval between measurements and a 5-min interval between raters

(*Morozumi et al., 2022*; *Walton et al., 2011*). The first and second measurements were used to assess intra-rater reliability, as they were conducted consecutively within the same session to minimize physiological variability (*Kottner et al., 2011*). The third measurement from each rater was used to assess inter-rater reliability, as this approach may help avoid potential learning or fatigue effects from earlier trials and may ensure standardized comparison timing between raters (*Kottner et al., 2011*).

To evaluate responsiveness using the anchor-based method, all pre- and post-intervention measurements were conducted by the beginner rater. A total of 36 additional participants were assessed immediately before the first physical therapy session and again one day after the final session. The most painful MTrP was identified through palpation, and its location was marked on a transparent sheet aligned with anatomical landmarks (acromion and C7 spinous process) to ensure consistent repositioning during the post-treatment assessment. All outcome measures were collected at both time points, and a single value was used for analysis. Additionally, the GRC was completed by each participant one day after the final treatment session.

## Statistical analysis

All data analyses and visualizations were performed using R software (Version 1.3.1093, RStudio PBC, 2009–2020). The Shapiro-Wilk test was used to assess the normality of all continuous variables.

Reliability in this study was evaluated from both relative and absolute perspectives. Relative reliability, which reflects the consistency with which individuals can be distinguished from one another despite measurement error, was quantified using the intraclass correlation coefficient (ICC), specifically the single-rating, absolute-agreement, two-way mixed-effects model ($ICC_{3,1}$). For this analysis, the first and second measurements were used to assess intra-rater reliability, while the third measurement was used for inter-rater reliability. Data were grouped according to the measurement sequence (1st, 2nd, and 3rd measurements). ICC values were interpreted as follows: <0.5 = poor, 0.5–0.75 = moderate, 0.75–0.9 = good, and >0.9 = excellent reliability (*Koo & Li, 2016*). Absolute reliability, often described as repeatability or agreement, reflects the precision of repeated measurements within the same participants. In this study, absolute reliability was assessed by evaluating measurement agreement using Bland–Altman analysis, which examines systematic bias, including both fixed and proportional bias. Limits of agreement (LoA) were calculated as the mean difference ±1.96 times the standard deviation (SD) of the differences, with 95% confidence intervals (CI) computed for both the upper and lower bounds (UB and LB) (*Koo & Li, 2016*). Additionally, the mean absolute percentage change was calculated to evaluate the magnitude of agreement in both TH and PPT measurements. For intra-rater agreement, the percentage change was calculated as: (second measurement – first measurement)/first measurement × 100. For inter-rater agreement, the percentage change was calculated as: (beginner's third measurement – expert's third measurement)/expert's third measurement × 100.

Responsiveness was evaluated using both distribution-based and anchor-based methods. Distribution-based methods: Paired t-tests were used to compare baseline and

post-treatment values. Effect size (ES) was calculated using Cohen's d as ES = (Post–Pre)/ SD of the differences. The standardized response mean (SRM) was calculated as SRM = Mean change/SD of change. The standard error of measurement (SEM) was estimated as SEM = SD × √(1–ICC), and the minimal detectable change at the 95% confidence level (MDC$_{95}$) was calculated as MDC$_{95}$ = SEM × 1.96 × √2. Anchor-based methods: Pearson's correlation was used to assess the correlation between global rating of change (GRC) and changes in TH and PPT, with a significance threshold set at r ≥ 0.30 (*Revicki et al., 2008*). The minimal clinically important difference (MCID) was determined using receiver operating characteristic (ROC) curves, with the area under the curve (AUC) used to assess how well outcome measures, such as TH and PPT, distinguished between patients reporting meaningful improvement (GRC ≥ 2) and those who did not (GRC < 2) (*Kamper, Maher & Mackay, 2009*).

Additional Pearson's correlation analyses were performed to explore correlation between changes in TH, PPT, and other clinical outcomes (VAS and NDI), with a *p*-value of less than 0.05 considered statistically significant.

# RESULTS

## Intra-rater reliability and inter-rater reliability for TH and PPT measurements

Participants with MTrPs in the UT muscle (*n* = 24) were predominantly female (21; 87.5%), and three males (12.5%). The mean age was 26.4 ± 5.39 years, and the BMI was 22.69 ± 3.93 kg/m². The most painful side was most frequently reported on the right side of the body 13 (54.16%).

The intra-rater reliability for both the expert and beginner raters was excellent for both TH and PPT measurements. For the expert, the ICC$_{3,1}$ value were 0.97 (95% CI [0.93–0.99]) for TH and 0.92 (95% CI [0.81–0.96]) for PPT. Similarly, the beginner showed ICC$_{3,1}$ value of 0.95 (95% CI [0.88–0.98]) for TH and 0.92 (95% CI [0.83–0.97]) for PPT. However, the inter-rater reliability between the expert and beginner was only moderate, with ICC$_{3,1}$ value of 0.60 for both TH (95% CI [0.27–0.80]) and PPT (95% CI [0.27–0.81]), suggesting greater variability between raters (Table 1).

Table 2 summarizes the fixed and proportional bias in TH and PPT measurements based on Bland–Altman analysis. For TH, fixed bias was found only in the expert's intra-rater agreement (95% CI [−2.75 to −0.16%]). For PPT, fixed bias was observed in the beginner's intra-rater agreement (95% CI [0.01–0.15 kg/cm²]), and inter-rater agreement (95% CI [−0.53 to −0.14%]). Proportional bias was also present in both TH and PPT measurements across all comparisons (*p* < 0.01).

Figures 2 and 3 present Bland–Altman plots illustrating the intra- and inter-rater agreement of TH and PPT measurements. For TH, the intra-rater agreement showed small mean differences between repeated measurements: −1.46 ± 3.23% for the expert and 1.22 ± 3.76% for the beginner. The LoA ranged from −7.79 to 4.87% and −6.15 to 8.60%, respectively. However, inter-rater agreement showed greater variability, with a mean difference of 1.57 ± 10.01% and LoA from −18.06 to 21.20%. For PPT, intra-rater

**Table 1 Reliability of tissue hardness (TH) and pressure pain threshold (PPT) measurements in 24 participants with myofascial trigger points (MTrPs) in upper trapezius (UT) muscle.**

| Raters | TH measurements Means ± SD | | | Intra-rater (ICC$_{3,1}$) 95% CI | Inter-rater (ICC$_{3,1}$) 95% CI | PPT measurements Means ± SD | | | Intra-rater (ICC$_{3,1}$) 95% CI | Inter-rater (ICC$_{3,1}$) 95% CI |
|---|---|---|---|---|---|---|---|---|---|---|
| | First | Second | Third | | | First | Second | Third | | |
| Expert | 44.86 ± 13.64 | 43.40 ± 12.37 | 44.92 ± 11.96 | 0.97 [0.93–0.99] | 0.60 [0.27–0.80] | 1.87 ± 0.54 | 1.84 ± 0.59 | 1.82 ± 0.54 | 0.92 [0.81–0.96] | 0.60 [0.27–0.81] |
| Beginner | 47.43 ± 12.58 | 48.64 ± 10.67 | 46.49 ± 10.37 | 0.95 [0.88–0.98] | | 1.40 ± 0.42 | 1.49 ± 0.46 | 1.48 ± 0.55 | 0.92 [0.83–0.97] | |

**Note:**
Values are presented as means ± standard deviations (SD). Tissue hardness (TH) is reported as a percentage (%), and pressure pain threshold (PPT) is reported in kilograms per square centimeter (kg/cm$^2$). Intra-rater reliability was assessed using the first and second measurements from each rater, while inter-rater reliability was calculated using the third measurement from each rater. ICC$_{3,1}$ = Intraclass correlation coefficient (two-way mixed-effects model, absolute agreement) with 95% confidence interval (CI).

**Table 2 Determination of fixed bias and proportional bias using Bland–Altman analysis for tissue hardness (TH) and pressure pain threshold (PPT) measurements in 24 participants with myofascial trigger points (MTrPs) in upper trapezius (UT) muscle.**

| | TH | | | | | PPT | | | | |
|---|---|---|---|---|---|---|---|---|---|---|
| | Fixed bias | | Proportional bias | | | Fixed bias | | Proportional bias | | |
| | 95% CI | | t value | p-value | | 95% CI | | t value | p-value | |
| Intra-rater (Expert) | [−2.75 to −0.16] | Presence | 0.88 | <0.01* | Presence | [−0.12 to 0.06] | Absence | 0.99 | <0.01* | Presence |
| Intra-rater (Beginner) | [−0.28 to 2.73] | Absence | 0.81 | <0.01* | Presence | [0.01 to 0.15] | Presence | 1.03 | <0.01* | Presence |
| Inter-rater | [−2.44 to 5.58] | Absence | 0.52 | <0.01* | Presence | [−0.53 to −0.14] | Presence | 0.62 | <0.01* | Presence |

**Note:**
Fixed bias was considered present when the 95% confidence interval (CI) of the mean difference did not include zero. Proportional bias was assessed using linear regression, with a statistically significant slope ($p < 0.01$) indicating its presence. Tissue hardness (TH) is reported as a percentage (%), and pressure pain threshold (PPT) is reported in kilograms per square centimeter (kg/cm$^2$). Asterisks (*) denote statistical significance at $p < 0.01$.

agreement was high for both raters, with minimal mean differences: −0.03 ± 0.23 kg/cm$^2$ for the expert and 0.08 ± 0.17 kg/cm$^2$ for the beginner. The corresponding LoA were −0.49 to 0.43 kg/cm$^2$ and −0.25 to 0.42 kg/cm$^2$, respectively. Inter-rater agreement also showed acceptable agreement, with a mean difference of −0.33 ± 0.49 kg/cm$^2$ and LoA ranging from −1.29 to 0.62 kg/cm$^2$. Additionally, the mean absolute percentage change for TH was 6.64 ± 5.16% (expert), 7.43 ± 8.09% (beginner), and 22.98 ± 23.62% (inter-rater). For PPT, the corresponding values were 9.13 ± 8.37%, 10.89 ± 11.96%, and 24.45 ± 19.53%, respectively.

## Responsiveness of TH and PPT measurements

An additional 36 participants with MTrPs in the UT muscle were recruited to receive PT treatment. The majority were female 28 (77.78%), with eight males (22.22%). The mean age was 26.0 ± 5.48 years, and the BMI was 22.47 ± 3.73 kg/m$^2$. The right side of the body was most frequently reported as the most painful side 22 (61.12%).

After six-session of PT treatment, both TH and PPT measurements demonstrated changes in the expected direction. The mean difference was −7.86% for TH (95% CI [−10.12 to −5.60%]; $p < 0.001$) and 0.20 kg/cm$^2$ for PPT (95% CI [0.04 to 0.36 kg/cm$^2$]; $p < 0.05$), indicating a reduction in the TH and an increase in the PPT. The ES were −1.18

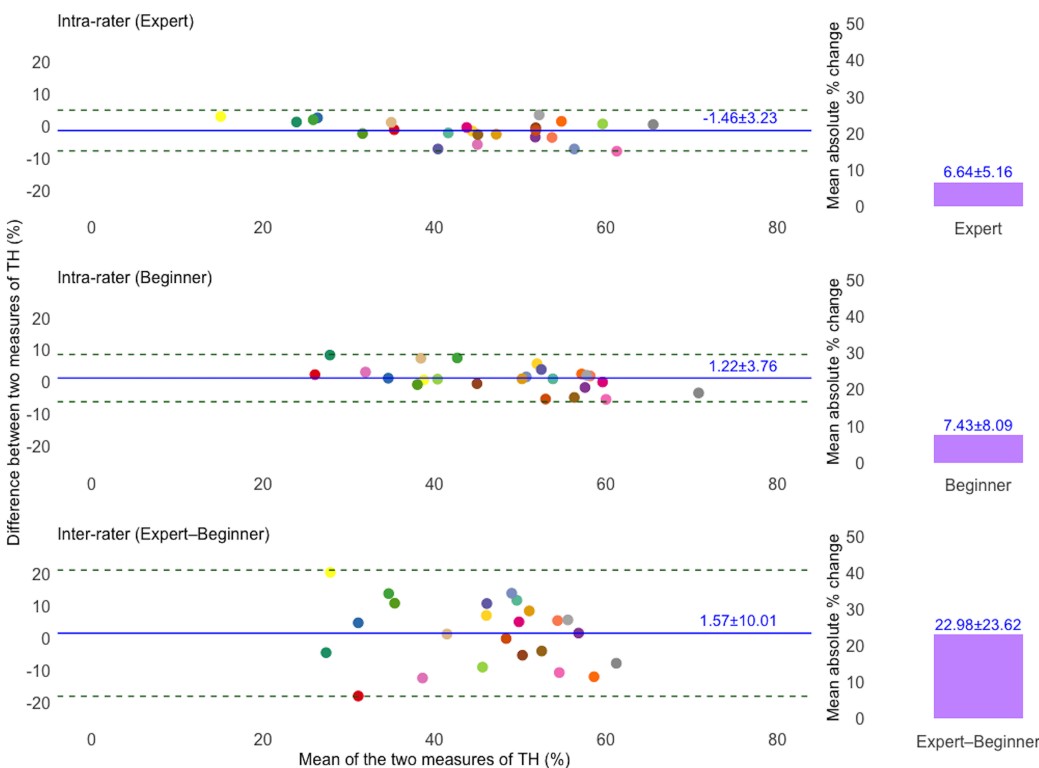

**Figure 2** **Bland–Altman plots illustrating intra- and inter-rater agreement of tissue hardness (TH) measurements for expert and beginner raters.** The solid blue line indicates the mean difference (bias) between two measurements, and dashed lines show the 95% limits of agreement (LoA). The unique colors of the data points refer to individual participants. Bar charts showing the mean absolute percentage (%) change in TH measurements. Top and middle plots represent intra-rater agreement, and the bottom plot represents inter-rater agreement.

(95% CI [−1.50 to −0.85%]) for TH and 0.42 (95% CI [0.09–0.74 kg/cm$^2$]) for PPT, with corresponding SRM of −1.18 and 0.42, respectively. The SEM was 2.66% for TH and 0.12 kg/cm$^2$ for PPT, while the MDC$_{95}$ was 7.37% for TH and 0.34 kg/cm$^2$ for PPT (Table 3).

For the anchor-based approach, Pearson's correlation between GRC and TH change was 0.22, and for PPT change was 0.03. The MCID was −7.43% for TH and 0.21 kg/cm$^2$ for PPT. The AUC was 0.97 for TH and 0.86 for PPT (Table 3).

## Correlation between changes in TH and PPT, and changes in clinical outcomes

Paired t-tests were conducted to compare clinical outcomes before and after the treatment. Results showed a significant reduction in VAS −3.01 cm (95% CI [−3.58 to −2.42 cm]; $p < 0.001$) and NDI −7.08 scores (95% CI [−9.49 to −4.68 scores]; $p < 0.001$).

Pearson's correlation analysis was performed to explore the correlations between changes in TH and PPT with changes in clinical outcomes, including VAS and NDI, following six-session of PT treatment. Changes in TH showed no significant correlation with any clinical outcomes. However, changes in PPT were negatively correlated with
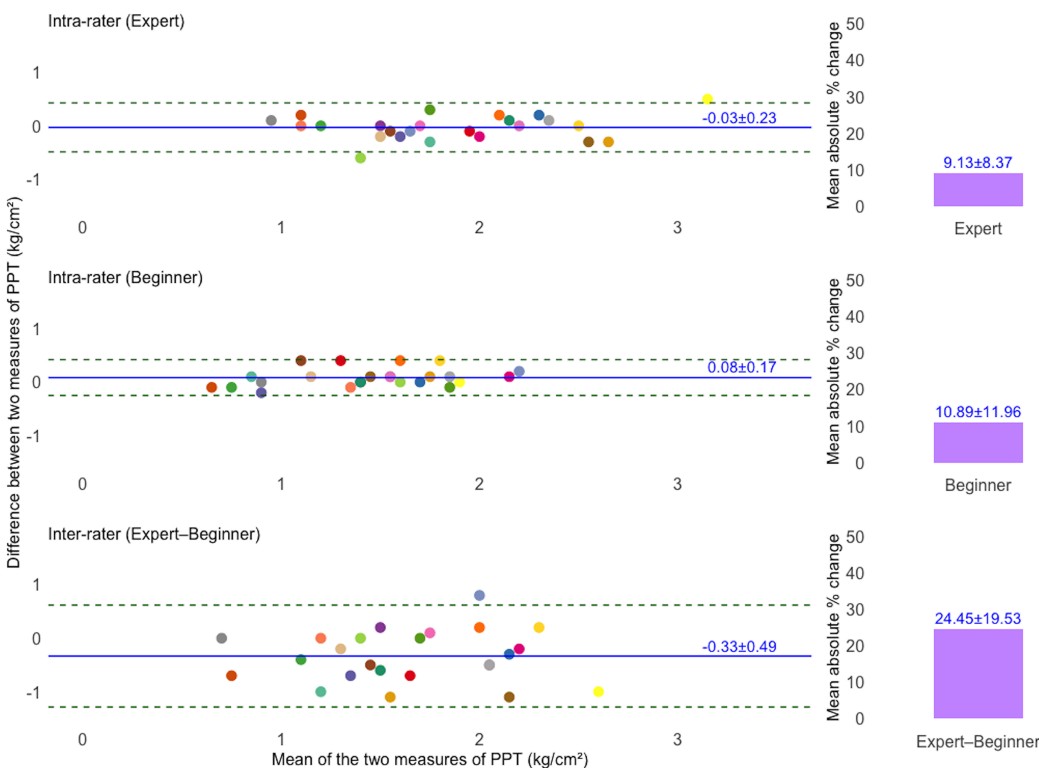

**Figure 3 Bland–Altman plots illustrating intra- and inter-rater agreement of pressure pain threshold (PPT) measurements.** The solid blue line indicates the mean difference (bias) between two measurements, and dashed lines show the 95% limits of agreement (LoA). The unique colors of the data points refer to individual participants. Bar charts showing the mean absolute percentage (%) change in PPT measurements. Top and middle plots represent intra-rater agreement, and the bottom plot represents inter-rater agreement.

**Table 3 Responsiveness of tissue hardness (TH) and pressure pain threshold (PPT) measurements in participants with myofascial trigger points (MTrPs) in upper trapezius (UT) muscle.**

| Variable | TH (%) | PPT (kg/cm$^2$) |
|---|---|---|
| **Distribution-based methods** | | |
| Mean difference[b] | −7.86 ± 6.68 (−10.12 to −5.60)** | 0.20 ± 0.47 (0.04 to 0.36)* |
| ES[b] | −1.18 (−1.50 to −0.85) | 0.42 (0.09 to 0.74) |
| SRM[b] | −1.18 | 0.42 |
| SEM[a] | 2.66 | 0.12 |
| MDC$_{95}$[a] | 7.37 | 0.34 |
| Anchor-based methods | | |
| Pearson's correlation[b] | 0.22 | 0.03 |
| MCID[b] | −7.43 | 0.21 |
| AUC[b] | 0.97 | 0.86 |

**Note:**
(a) Data were calculated on the sample ($n = 24$); (b) data were calculated on the sample ($n = 36$); 95% confidence intervals are shown in parentheses; ES, effect size; SRM, standardized response mean; SEM, standard error of the measurements; MDC$_{95}$, minimum detectable change at 95% confidence interval; MCID, minimal clinically important difference; AUC, area under the curve; (%), percentage; kg/cm$^2$, kilograms per square centimeter. Asterisks (*) indicates statistical significance ($p < 0.05$); (**) indicates statistical significance ($p < 0.001$).

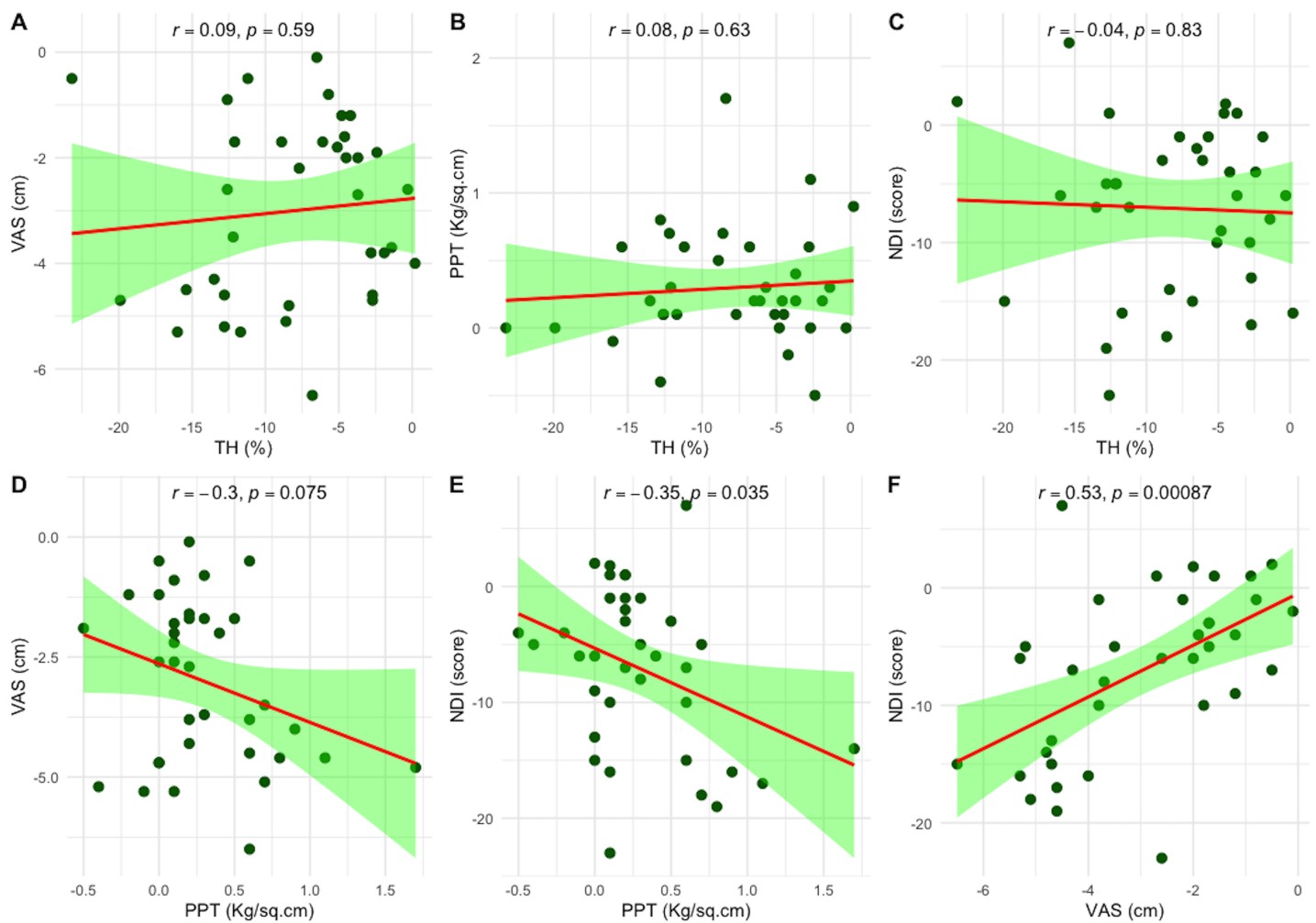

**Figure 4 Scatter plots showing the correlations between changes in objective measures—tissue hardness (TH) and pressure pain threshold (PPT)—and clinical outcomes following six sessions of physical therapy.** (A–C) Correlations between changes in TH and changes in VAS, PPT, and NDI, respectively. (D–F) Correlations between changes in PPT and changes in VAS and NDI, and between VAS and NDI, respectively. Each plot includes Pearson's correlation coefficient (r); statistical significance was set at $p < 0.05$. Pain intensity was measured using the visual analog scale (VAS), and disability using the Neck Disability Index (NDI).

changes in NDI ($r = -0.35$, $p = 0.035$). Additionally, the change in VAS was positively correlated with the change in NDI ($r = 0.53$, $p < 0.001$), as illustrated in Fig. 4.

## DISCUSSION

This study provides evidence that both TH and PPT measurements in a side-lying position are reliable when performed by the same rater. However, consistency decreases when different raters assess patients. This highlights a significant challenge in clinical practice, where multiple clinicians may assess the same patient. Ensuring consistent and reliable measurements across raters is therefore essential for accurate assessment and treatment evaluation.

Excellent intra-rater reliability ($ICC_{3,1} > 0.90$) was found for both TH and PPT measurements, which is consistent with previous studies (*Damapong et al., 2015*; *Waller*

*et al., 2015*). However, inter-rater reliability was moderate (ICC$_{3,1}$ = 0.60) for both measurements. This discrepancy was further supported by the mean absolute percentage changes, which indicated greater variability in inter-rater agreement compared to intra-rater agreement. This contrasts with earlier studies reporting excellent inter-rater reliability (ICC = 0.92) for PPT when measurements were taken at a standardized location—such as the midpoint of the UT muscle between the C7 spinous process and the lateral acromion (*Waller et al., 2015*). In contrast, our protocol assessed measurement reliability based on the location of the most painful MTrP across all regions of the UT muscle. This approach is comparable to a previous study in which PPT was measured at symptomatic sites within the upper fibers of the UT muscle while participants were seated, yielding excellent intra-rater reliability (ICC = 0.96) but only good inter-rater reliability (ICC = 0.81) (*Walton et al., 2011*). Our approach reflects clinical practice more closely but may introduce variability in inter-rater assessments. Additionally, the moderate inter-rater reliability may partly reflect the sample's homogeneity, as most participants were young adults, which can reduce between-subject variance and lead to lower ICC values.

Our finding highlights the impact of rater-related variation and further supports the importance of standardization in both clinical and research contexts. When feasible, follow-up assessments should be conducted by the same rater to take advantage of the superior intra-rater reliability. In clinical settings with multiple raters, implementing the SEM monitoring system could help identify inconsistencies before they compromise data integrity. Additionally, to minimize random error and improve measurement consistency, we recommend averaging multiple measurements rather than relying on a single value when using this device.

The Bland–Altman analysis revealed both fixed and proportional biases in measurements. Fixed biases were identified in the expert's TH measurements and the beginner's PPT measurements, consistent with previous research (*Morozumi et al., 2022*; *Zicarelli et al., 2021*), likely due to variations in technique and experience (*Morozumi et al., 2022*). Specifically, the expert consistently recorded lower TH values in the second measurement, indicating a systematic tendency to measure lower in subsequent tests. Conversely, the beginner recorded higher PPT values in the second measurement, indicating a systematic tendency to measure higher.

Proportional bias was observed in both TH and PPT measurements, indicating that measurement accuracy decreased as the values increased—suggesting that higher threshold values are more prone to error. This finding differs from previous studies, which did not report proportional bias, even though TH measurements were performed using: (1) three different devices—one operated *via* damped oscillations and two required manual pressure application by the rater; (2) different muscle conditions, including relaxed and stretched positions; and (3) raters with varying levels of experience, including both experts and beginners. One possible explanation for this discrepancy may lie in the protocol used in our study, which positioned participants in a side-lying posture. This positioning required the rater to apply force perpendicular to the direction of gravity, whereas previous studies applied pressure aligned with the gravitational axis. This orientation may have made it more challenging to maintain a consistent application rate of 1 kg/cm$^2$ per second

and ensure full contact of the plastic plate with the tissue surface throughout the measurement process (*Morozumi et al., 2010, 2022*; *Walton et al., 2011*; *Waller et al., 2015*). Particularly in the side-lying position, the use of an adjustable bed may be more appropriate than requiring the examiner to bend their knees during measurement. This setup may facilitate the application of perpendicular force and ensure full contact between the device disk and the target area.

The presence of both fixed and proportional biases indicates that measurement variability may persist even when standardized procedures are followed—particularly between raters. This emphasizes the need for improved measurement protocols, including ensuring perpendicular force application, maintaining a consistent pressure application rate, achieving full contact of the flat plastic disk with the tissue surface, and refining device design to enhance control over the applied force—such as the use of automatic constant-velocity or force-controlled algometers (*Melia et al., 2015*).

Following six physical therapy sessions, a significant reduction in TH was observed −7.86% (95% CI [−10.12 to −5.60%], $p < 0.001$), along with a moderate increase in PPT 0.20 kg/cm$^2$ (95% CI [0.04–0.36 kg/cm$^2$], $p < 0.001$). These findings are consistent with prior studies that utilized the THA device to evaluate the effects of therapeutic interventions on MTrPs in the UT muscle. Interventions such as self-massage using the Wilai massage stick (*Wamontree, Kanchanakhan & Eungpinichpong, 2016*), court-type traditional Thai massage (*Damapong et al., 2015*), and the squeezed technique of Thai massage for neck strain (*Panngooluema & Eungpinichpong, 2021*) have demonstrated reductions in TH and improvements in PPT. Furthermore, devices with similar mechanisms to the THA have demonstrated sensitivity in detecting PPT changes in different contexts—for example, before and after computer work (*Park & Yoo, 2013*) and between groups using a neck support tying method *vs.* controls (*Yoo & Yoo, 2013*). While these findings support the responsiveness of the THA device in detecting treatment-related changes, it is noteworthy that none of these earlier studies reported or interpreted clinically meaningful thresholds, such as the MCID, limiting their clinical applicability.

A large effect size for TH (ES = −1.18) reflects strong clinical improvement, whereas the moderate effect size for PPT (ES = 0.42) indicates a smaller yet meaningful change. However, greater variability and a higher SEM for TH indicate lower precision compared to PPT. This raises questions about the ability of TH to accurately capture changes in muscle pain. The greater variability in TH measurements may result from extraneous factors, such as the composition of underlying tissues, including subcutaneous adipose tissue (ATT). Previous studies have shown that thicker ATT layers can interfere with TH assessments (*Chino & Takahashi, 2016*), complicating the measurement of true muscle-specific changes. This limitation underscores the need for more advanced techniques, such as magnetic resonance elastography (MRE), which may offer a more accurate evaluation of muscle-specific changes (*Chen et al., 2007, 2016*).

In the anchor-based analysis, there was a low correlation between patient-reported outcomes (GRC) and objective measures (TH and PPT). The low correlation (r = 0.22 for TH and r = 0.03 for PPT) indicates that patients' subjective perceptions of improvement do not always align with the physical changes measured by TH and PPT. This disconnect may

stem from psychological factors that influence patients' perceptions of improvement (*Yi et al., 2014*). Clinicians should consider this by integrating both subjective reports and objective measurements to gain a comprehensive understanding of treatment outcomes. Despite the low correlation between subjective and objective measures, the AUC analysis confirmed that TH is a highly accurate predictor of meaningful clinical improvement (AUC = 0.97), making it a valuable tool for assessing treatment success. PPT, with an AUC of 0.86, also performed well but was slightly less accurate than TH. These findings suggest that while both TH and PPT are useful, TH may be more sensitive in detecting meaningful changes.

The clinical relevance of our findings is further supported by the MCID, defined as the smallest change perceived as meaningful by patients: −7.43% for TH and 0.21 kg/cm$^2$ for PPT. In this study, the reduction in TH exceeded both the MDC$_{95}$ and MCID, indicating that the improvement was not only statistically significant but also clinically meaningful. In contrast, the change in PPT fell below both thresholds, suggesting a less substantial clinical effect. This may be attributed to the beginner rater, who demonstrated fixed bias in intra-rater agreement for PPT, but not for TH. Notably, the same device was previously used to assess TH and PPT in patients with long head of biceps brachii tendinopathy, where a 10–15% reduction in TH and a 1–1.5 kg/cm$^2$ increase in PPT were considered meaningful following dry needling treatment. However, that study did not define explicit MDC$_{95}$ or MCID values (*Chen et al., 2024*).

Another important finding was the negative correlation between PPT improvements and reductions in the NDI (r = −0.35). This indicates that as pain sensitivity improves, disability decreases, a result consistent with previous research (*Valera-Calero et al., 2021*). The positive correlation between changes in the VAS and NDI (r = 0.53) further supports the correlation between pain reduction and improved function.

The primary limitation of this study is its small sample size, which may limit the generalizability of the findings. Furthermore, we did not screen for ATT and control cervical spine positioning (*Snodgrass & Rhodes, 2012*), which could have influenced the TH and PPT measurements. Additionally, the responsiveness phase lacked a control group, limiting our ability to attribute changes in TH and PPT solely to the intervention. This introduces potential confounding factors such as natural recovery or placebo effects. Another important limitation relates to the large contact surface of the device, which may not have conformed equally across individuals with smaller body frames, potentially introducing variability in measurement accuracy. Future studies should consider using alternative tools with a smaller or well-defined contact area, where contact variability may be reduced. Although the device used in this study has been previously validated against electromyography and a commercial algometer (*Morozumi et al., 2010*), we acknowledge that it has not yet been directly validated against gold-standard instruments such as ultrasound elastography or myotonometry. Future studies should include larger sample sizes, more rigorous control of these factors, and conduct external validation to enhance the reliability and applicability of the results.

## CONCLUSIONS

This study shows that measuring TH and PPT using the THA device in a side-lying position demonstrates high reliability and responsiveness in patients with MTrPs in the UT muscle. With proper calibration and consistent rater practices, these measurements effectively capture clinically significant changes and support the evaluation of therapeutic interventions.

## ACKNOWLEDGEMENTS

The authors extend their gratitude to the patients who participated in this study.

### Funding

This work was supported by the Research Center in Back, Neck, Other Joint Pain and Human Performance (BNOJPH), Khon Kaen University (KKU), Khon Kaen, Thailand. The research facility at the Faculty of Associated Medical Sciences (AMS), Khon Kaen University, also provided institutional support. No external funding was received for this study. The funders had no role in study design, data collection and analysis, decision to publish, or preparation of the manuscript.

### Grant Disclosures

The following grant information was disclosed by the authors:
Research Center in Back, Neck, Other Joint Pain and Human Performance (BNOJPH).
Khon Kaen University (KKU), Khon Kaen, Thailand.
Faculty of Associated Medical Sciences (AMS).

### Competing Interests

The authors declare that they have no competing interests.

### Author Contributions

- Soukmisai Somphithak conceived and designed the experiments, performed the experiments, analyzed the data, prepared figures and/or tables, authored or reviewed drafts of the article, and approved the final draft.
- Uraiwan Chatchawan conceived and designed the experiments, authored or reviewed drafts of the article, and approved the final draft.
- Atipong Pimdee performed the experiments, authored or reviewed drafts of the article, and approved the final draft.
- Wiraphong Sucharit conceived and designed the experiments, performed the experiments, analyzed the data, prepared figures and/or tables, authored or reviewed drafts of the article, and approved the final draft.

### Human Ethics

The following information was supplied relating to ethical approvals (*i.e.*, approving body and any reference numbers):

This study was approved by the Research Ethics Committee of Khon Kaen University (KKU) in accordance with the Declaration of Helsinki (Reference: HE662181) and is registered in the Thai Clinical Trails Registry (TCTR20240618009).

## Data Availability

The raw measurements are available in the Supplemental File.

## Clinical Trial Registration

The following information was supplied regarding Clinical Trial registration:

Thai Clinical Trails Registry (TCTR20240618009).

## Supplemental Information

Supplemental information for this article can be found online at http://dx.doi.org/10.7717/peerj.19580#supplemental-information.

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
