# Peer review of "Reliability and responsiveness of a tissue hardness meter and algometer for measuring tissue hardness and pressure pain threshold in upper trapezius myofascial trigger points"

_PeerJ, doi:10.7717/peerj.19580_

## Round 0.1 · original submission · Major Revisions

Reviewers have substantial criticisms of the methodology. It is important to address all their comments if the manuscript is to be considered further.

Reviewer 1 ·

Basic reporting

These weaknesses in the reporting have been observed:

The study does not present a clear hypothesis or a well-defined research question. Without a solid theoretical foundation, the research appears exploratory rather than hypothesis-driven, which significantly reduces its scientific value.

There is no mention of pre-registration or adherence to standard scientific reporting guidelines such as CONSORT, STROBE, or PRISMA, which are essential for ensuring transparency and reproducibility.

Experimental design

• There is no clear control group (e.g., a sham treatment or waiting list group) to validate the treatment outcome. The lack of such a comparison makes it impossible to determine whether the observed effects are due to the intervention or other confounding factors.

• The study fails to implement blinding (e.g., single-blind or double-blind design), which raises concerns about experimenter bias in measurements. A minimum requirement for inter-rater reliability would have been to ensure that Evaluator 1 and Evaluator 2 were blinded to each other’s measurements from the same patients. Without this, the risk of observer bias remains high.

• The newly developed instrument is not validated against established measurement tools, such as ultrasound-elastography or myotonometry, which are currently used for assessing myofascial tissue stiffness. Without such an external comparison, it is unclear whether the device provides accurate and meaningful measurements in the described application.

Validity of the findings

• The large contact surface of the device (a flat plastic disc with a 10 cm diameter) may be appropriate for individuals with broad shoulders, where the contact area is known. However, in smaller individuals, the actual contact surface remains unknown, which significantly affects measurement accuracy. If the contact surface area (in cm²) was controlled, the methodology should explicitly describe how this was ensured.
If the contact surface was not standardized, the study should either:
1. Use alternative assessment tools with a smaller and better-defined contact area (like it is done in most other stiffness assessment methods)
2. Select larger muscles (e.g., gluteus maximus) where the contact variability is reduced.
3. Clearly discuss the severe limitations of the chosen measurement approach.

Additional comments

The study presents an interesting attempt to integrate tissue hardness measurement and algometry into a single device, and the authors provide detailed descriptions of their measurement procedures. If refined and validated with a more rigorous methodology, this approach could contribute to improving objective pain and tissue hardness assessments in clinical and research settings.

.However, the study lacks sufficient scientific rigor to be deemed acceptable for publication. Several critical weaknesses (explained above) undermine its validity and reliability.

While the concept of integrating a tissue hardness meter and an algometer is intriguing, the current study does not provide sufficient evidence to support its reliability and validity.

·

Basic reporting

Figure and Table Descriptions: While figures and tables are generally well-labeled and relevant, some may benefit from more detailed descriptions directly in the text to ensure clarity without needing to refer back to the figures or tables themselves.
Table 1 (Fixed and Proportional Bias):
The presence of both fixed and proportional biases in TH and PPT measurements, particularly across different raters, might impact the generalizability and consistency of these measurements. This information should be considered when interpreting the results and could be a point for discussion on improving measurement techniques.

Experimental design

Detailed Methodology: Some aspects of the methodology, such as specific calibration procedures for equipment or detailed descriptions of the statistical methods used, could be expanded for enhanced replicability.

Validity of the findings

1. Interpretation of Bias: The discussion of fixed and proportional biases is critical, especially given the moderate inter-rater reliability found. The manuscript would benefit from a deeper exploration of how these biases might impact the clinical application of the findings.
2. Broader Implications: Expanding the discussion to consider broader implications for clinical practice, and possibly suggesting specific protocols or training that could standardize measurements across different raters, would provide valuable context for the reader.

Additional comments

The manuscript is a valuable contribution to the field of physical therapy and rehabilitation. The study's design and execution are generally robust, but there are areas where additional details could enhance understanding and application of the research findings. Further clarification of methods and a more thorough exploration of the implications of rater variability and measurement bias would strengthen the manuscript.

·

Basic reporting

The paper is generally well-written but lacks a deeper explanation of the developed side-lying protocol. Expanding on this protocol would enhance the reader's understanding and improve the paper's contribution to the field.
The authors provide strong citations for prior work on algometry reliability studies; however, the paper fails to contextualize these references by comparing their reported values to the present study's findings. Including this information would provide better insight into the significance and implications of the results.
The tables and figures require improved formatting for clarity and professionalism. The text size varies inconsistently, spacing is uneven, and labeling is insufficient. I recommend centering text in tables for better readability and revisiting figure captions to ensure they are clear and informative.
The authors should elaborate on the measure of reliability used. It is unclear whether the focus is on accuracy, precision, or repeatability. While the authors write as if they are testing repeatability, the results suggest they are instead assessing precision. Adding a dedicated section that defines reliability in the study's context would improve clarity.

Experimental design

This paper describes a study designed to improve the reliability of skin state measurements by developing a side-lying protocol. The study involves two experiments:
The first experiment measures the repeatability of tissue hardness and pain pressure threshold across multiple measurements and between an expert and an amateur. Intra-repeatability is assessed using the difference between the first and second results, while inter-repeatability is assessed using the difference between the expert's and amateur's third result. However, the difference evaluation method lacks clear justification, and the absence of contextual data calls into question the validity of the reported results.
The second experiment measures tissue hardness and pain pressure threshold to assess skin state before and after six physical therapy sessions. While this experiment is valuable, it lacks a detailed description of the measurement protocol and fails to place the results in the context of previous studies. Clarifying the magnitude of observed changes and their significance would improve this section.
To strengthen the paper, the authors should provide a more detailed explanation of their results in the context of existing literature, expand upon the developed side-lying protocol, and refine their reliability/repeatability analysis.
The authors should also report the accuracy and precision of the measurement devices used (Line 111). Understanding the reliability of these devices would help readers differentiate between variance caused by device limitations and variance resulting from testing conditions or protocol design.
Details about the pre- and post-intervention measurement protocol are missing. The authors should clarify whether an expert administered these evaluations, how many evaluations were conducted, and when these measurements took place (e.g., immediately before the first session, immediately after the final session, etc.).
While the development of a side-lying protocol is valuable, it is insufficiently described. The authors should dedicate a section to detailing the steps of the protocol for replication purposes. Expanding Figure 1 to outline the procedure, identify bracer placement, and describe the participant's position would be helpful.

Validity of the findings

The first experiment aims to assess the reliability of tissue hardness and pain pressure threshold measurements across multiple tests and for trained vs. untrained administrators.
To validate the reported TH and PPT values, the authors should compare their findings with similar studies, such as those by Won-gyu Yoo and Walaa Abu Taleb. Discussing how these results align (or differ) would strengthen the paper's conclusions and contextualize the study's contributions.
Figures 2 and 3 would benefit from including absolute values for participant measurements. For example, in Figure 2, showing that Participant 1's TH values are ~37 during the first two WS measurements provides essential context for understanding the observed differences. Including percentage change values may also enhance the interpretation of consistency results.
The comparison method for intra- and inter-reliability lacks clear reasoning. The current method compares the first and second measurement for intra-reliability while only considering the third measurement for inter-reliability. A more representative approach would involve averaging all three measurements for intra-reliability and comparing these averages for inter-reliability. Alternatively, reporting sequential differences for each participant could provide a more comprehensive view of measurement variability.
The paper's use of ICC in reliability evaluation requires further clarification. The values used for evaluation are vaguely described (Line 141). The authors should specify what data points were included (test results for each administrator, differences between test results, or differences between administrators) and how participant data were grouped (e.g., by test number, administrator, or participant). Additionally, the authors should discuss the potential for low ICC values resulting from homogeneity among participants. Expanding this discussion would strengthen the interpretation of the ICC results.
In the second experiment, which measures TH and PPT values before and after six physical therapy sessions, there is no mention of VAS (Visual Analog Scale) scores. Reporting this data would be helpful, especially since the authors note a decrease in pain intensity across participants. Given that reduced pain intensity is a desired therapy outcome, including these scores would reinforce the paper's findings.

Additional comments

The inclusion of the TH and PPT datasets is great; however, the dataset's labeling is unclear. Adding a readme document that defines each label's meaning would improve usability. Furthermore, the authors should ensure that units are provided for each column.
Figure 1 should be improved by removing excess black lines along the right and bottom edges. Additionally, the text above the figure should be reformatted to include a distinct title and clearer body text with reduced font size.
Line 253's statement that "Our findings confirm that both pain sensitivity and pain intensity significantly impact disability" feels unfounded. The paper does not cover disability, making this statement seem misplaced. Removing or rephrasing this statement to align with the study's scope would improve coherence.
The caption for Figure 3 appears incomplete, starting with "shows" in lowercase. Expanding the caption to provide clear context would improve clarity.
The labels for solid and dotted lines in Figures 2 and 3 are too cluttered. Moving these labels to the side would improve readability. Additionally, the empty white space in these figures could be better utilized by including side charts displaying measurement magnitudes for each participant. This addition would provide useful context for the reported differences.

---

## Round 0.2 · accepted · Accept

Dear Authors

Thank you for addressing all the concerns raised by the referees. Congratulations on the acceptance of your work!

Reviewer 1 ·

Basic reporting

no comment

Experimental design

no comment

Validity of the findings

no comment

Additional comments

The previously described weak points have been sufficiently addressed.

·

Basic reporting

All reviewer recommendations were addressed with professionalism and scientific rigor. The revised manuscript demonstrates clear improvements in methodological clarity, statistical transparency, and clinical applicability. I recommend acceptance pending any final formatting or language refinements by the editorial team.

Experimental design

-

Validity of the findings

-